# A switch in surface polymer biogenesis triggers growth-phase-dependent and antibiotic-induced bacteriolysis

Josué Flores-Kim[1], Genevieve S Dobihal[1], Andrew Fenton[1,2], David Z Rudner[1]*, Thomas G Bernhardt[1,3]*

[1]Department of Microbiology, Harvard Medical School, Boston, United States; [2]The Florey Institute, Molecular Biology Biotechnology, University of Sheffield, Sheffield, United Kingdom; [3]Howard Hughes Medical Institute, Boston, United States

**Abstract** Penicillin and related antibiotics disrupt cell wall synthesis to induce bacteriolysis. Lysis in response to these drugs requires the activity of cell wall hydrolases called autolysins, but how penicillins misactivate these deadly enzymes has long remained unclear. Here, we show that alterations in surface polymers called teichoic acids (TAs) play a key role in penicillin-induced lysis of the Gram-positive pathogen *Streptococcus pneumoniae* (*Sp*). We find that during exponential growth, *Sp* cells primarily produce lipid-anchored TAs called lipoteichoic acids (LTAs) that bind and sequester the major autolysin LytA. However, penicillin-treatment or prolonged stationary phase growth triggers the degradation of a key LTA synthase, causing a switch to the production of wall-anchored TAs (WTAs). This change allows LytA to associate with and degrade its cell wall substrate, thus promoting osmotic lysis. Similar changes in surface polymer assembly may underlie the mechanism of antibiotic- and/or growth phase-induced lysis for other important Gram-positive pathogens.
DOI: https://doi.org/10.7554/eLife.44912.001

*For correspondence:
david_rudner@hms.harvard.edu (DZR);
thomas_bernhardt@hms.harvard.edu (TGB)

**Competing interests:** The authors declare that no competing interests exist.

## Introduction

Penicillin and related beta-lactams are some of our oldest and most effective antibiotics. These drugs disrupt the cell wall biogenesis pathway in bacterial cells and in doing so elicit explosive lysis (*Park, 1964*; *Park and Strominger, 1957*). Despite the long history of their use in the clinic, we still know relatively little about how beta-lactams trigger this catastrophic event. Understanding the mechanisms responsible for this lethal activity has the potential to provide fundamental new insights into the cell wall assembly process and to reveal novel ways of inducing bacteriolysis for future therapeutic development.

The main targets of the beta-lactams are cell wall synthases called penicillin-binding proteins (PBPs) (*Strominger and Tipper, 1965*; *Tipper and Strominger, 1965*), of which there are two varieties: class A (aPBPs) and class B (bPBPs) (*Goffin and Ghuysen, 1998*; *Sauvage et al., 2008*). These enzymes build the peptidoglycan (PG) matrix that surrounds most bacterial cells and protects them from osmotic rupture. The aPBPs are bifunctional and possess PG glycosyltransferase (PGT) and transpeptidase (TP) activity to polymerize the glycan strands of PG and form the inter-strand crosslinks of the network, respectively. The bPBPs, on the other hand, only possess TP activity. They work in concert with a second class of PG polymerases called the SEDS (shape, elongation, division, and sporulation) family proteins to synthesize and crosslink new cell wall material (*Cho et al., 2016*; *Emami et al., 2017*; *Meeske et al., 2016*; *Sjodt et al., 2018*; *Taguchi et al., 2019*).

Beta-lactams covalently modify the TP active site of the PBPs to block the PG crosslinking activity of the aPBPs and SEDS-bPBP complexes (*Cho et al., 2016*; *Tipper and Strominger, 1965*). The first

models of the lytic mechanism following TP inhibition by these drugs proposed that lysis resulted from the gradual weakening of the wall as it was made with fewer and fewer crosslinks (*Park and Strominger, 1957*). However, pioneering studies from Tomasz and co-workers in the Gram-positive pathogen *Streptococcus pneumoniae* (*Sp*) changed this view by showing that lysis in this organism was dependent on the activity of a PG cleaving enzyme called LytA (*Tomasz et al., 1970*; *Tomasz and Waks, 1975*). Beta-lactam-induced lysis in other bacteria was subsequently shown to also be dependent on factors with PG processing activity (*Cho et al., 2014*; *Chung et al., 2009*; *Heidrich et al., 2001*; *Heidrich et al., 2002*; *Tipper and Strominger, 1965*; *Uehara et al., 2009*). Thus, these PG hydrolases are often referred to as autolysins, and from genome sequencing data we now know that most bacteria encode a large variety of enzymes capable of breaking bonds in the PG network (*Uehara and Bernhardt, 2011*; *Vollmer et al., 2008*).

In normally growing bacteria, PG hydrolases are thought to participate in a number of critical cellular processes. Their activity is important to create space in the existing wall matrix to allow for the insertion of new material during cell elongation (*Bisicchia et al., 2007*; *Carballido-López et al., 2006*; *Meisner et al., 2013*; *Sycuro et al., 2010*; *Vollmer et al., 2008*). They also remodel the cell wall formed at the division site (septal PG) to shape the new poles and promote daughter cell separation (*De Las Rivas et al., 2002*; *Fan, 1970*; *Heidrich et al., 2001*; *Hett et al., 2008*; *Lominski et al., 1958*). Given their potential to induce cell lysis, it has long been appreciated that bacteria must possess robust mechanisms to control when and where PG hydrolases are activated to cut bonds in the PG network (*Uehara and Bernhardt, 2011*; *Vollmer et al., 2008*). However, surprisingly little is known about the regulatory systems that control these potentially deadly enzymes or how they are subverted by antibiotics like beta-lactams to lyse bacterial cells.

To better understand PG hydrolase regulation and penicillin-induced lysis, we used *Sp* as a model system. It has the advantage of requiring a single PG hydrolase called LytA for lysis-induction (*Figure 1A* and *Figure 1—figure supplement 1*) (*Tomasz et al., 1970*; *Tomasz and Waks, 1975*). The problem is therefore more genetically tractable in *Sp* than in other model organisms where multiple PG hydrolases are implicated in lysis-induction (*Heidrich et al., 2001*; *Uehara et al., 2009*;

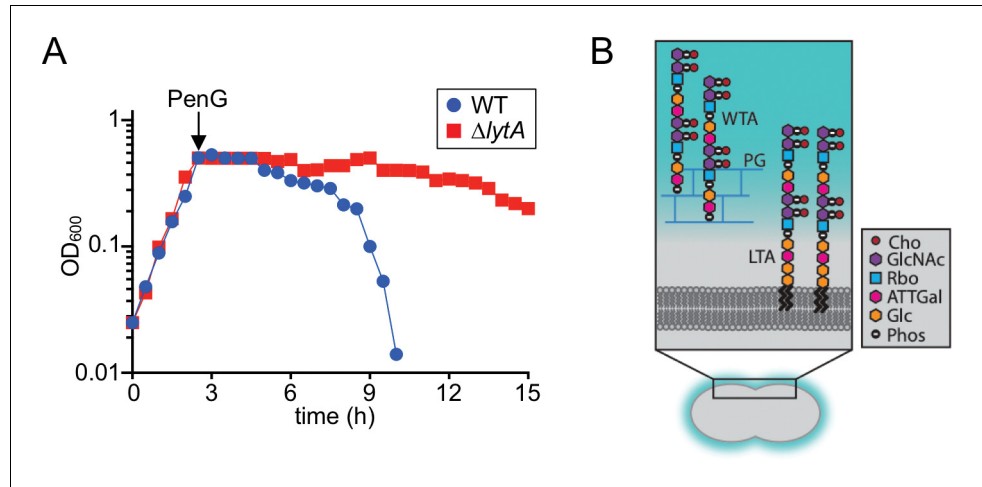

**Figure 1.** Beta-lactam induced lysis of *Sp* and overview of its cell envelope. (**A**) The indicated strains were grown in THY at 37 °C in 5% $CO_2$. At an $OD_{600}$ of ~0.5, they were challenged with penicillin G (PenG) (0.5 µg/ml final). Growth was monitored every 30 min for 15 hr. (**B**) Schematic diagram of the cell envelope of *Sp*. The cell wall peptidoglycan (PG) (blue) contains Wall Teichoic Acid (WTA) polymers and the lipid bilayer contains Lipoteichoic Acid (LTA). The constituents of the repeating unit in LTAs and WTAs are indicated; Cho, choline; GlcNac, *N*-acetylglucosamine; Rbo, ribitol; ATTGal, 2-acetamido-4-amino-2,4,6-trideoxygalactose; Glc, glucose; Phos, phosphate. The following figure supplement is available for *Figure 1*.

DOI: https://doi.org/10.7554/eLife.44912.002

The following figure supplement is available for figure 1:

**Figure supplement 1.** LytA levels remain constant before and at the onset of growth-phase-dependent autolysis.

DOI: https://doi.org/10.7554/eLife.44912.003

*Vollmer et al., 2008*). Another benefit of *Sp* is its propensity to lyse following prolonged growth in stationary phase (*Fernebro et al., 2004*; *Mellroth et al., 2012*; *Tomasz et al., 1970*; *Tomasz and Waks, 1975*). Like penicillin-induced lysis, autolysis in stationary phase is LytA-dependent (*Fernebro et al., 2004*; *Mellroth et al., 2012*; *Tomasz et al., 1970*; *Tomasz and Waks, 1975*). This property of *Sp* cells enabled us to develop a genetic screen for LytA regulators. The screen revealed a key role for surface polymers called teichoic acids (TAs) in controlling LytA activity. TAs are major constituents of the cell surface in Gram-positive bacteria and are either lipid-anchored (lipoteichoic acids, LTAs) or wall-anchored (wall teichoic acids, WTAs) (*Figure 1B*) (*Brown et al., 2013*; *Percy and Gründling, 2014*). Our results indicate that *Sp* cells primarily produce LTAs during normal exponential growth, which bind and sequester LytA. However, entry into stationary phase and penicillin-treatment were both found to trigger the degradation of the *Sp* LTA synthase, causing a switch to the production of WTAs. This change allows LytA to associate with and degrade its cell wall substrate, thus promoting osmotic lysis. We propose that changes in surface polymer assembly may similarly underlie the mechanism of antibiotic-induced lysis for a number of other important Gram-positive pathogens.

## Results

### Identification of TacL as a potential LytA control factor

Previous studies indicated that LytA protein levels remain constant during growth, and we have confirmed this result (*Figure 1—figure supplement 1*) (*Fernebro et al., 2004*; *Henriques Normark and Normark, 2002*; *Mellroth et al., 2012*). Based on this observation, we hypothesized that LytA activity is negatively regulated during normal exponential growth by an inhibitory factor(s). In this scenario, inhibition would be relieved upon entry into stationary phase or exposure to cell wall synthesis inhibitors triggering lysis. If correct, this hypothesis predicts that the putative LytA inhibitory factor(s) should be essential for growth in wild type (LytA$^+$) cells but become non-essential in cells lacking LytA ($\Delta lytA$). Therefore, to identify the potential LytA inhibitor(s), we used transposon sequencing (Tn-Seq) to screen for *Sp* genes displaying the expected pattern of essentiality/non-essentiality. Transposon libraries were prepared in a wild-type strain D39 without its capsule (WT) and a derivative deleted for *lytA* ($\Delta lytA$) (*Fenton et al., 2016*; *Land and Winkler, 2011*). When the insertion profiles were compared, we found that the gene *tacL* (SPD_1672) was virtually devoid of insertions in the WT library, but readily inactivated by insertions in the $\Delta lytA$ library (*Figure 2A*). To validate the Tn-Seq results, we constructed a TacL-depletion strain in which the sole copy of *tacL* was placed under control of a zinc-regulated promoter ($\Delta tacL$ P$_{Zn}$-*tacL*) (*Eberhardt et al., 2009*). When TacL was produced (+Zn), cells were viable regardless of their LytA status (*Figure 2B*). However, when TacL was depleted (-Zn), viability was severely compromised only in cells producing LytA (*Figure 2B*). Furthermore, TacL depletion during growth in liquid medium caused premature LytA-dependent autolysis in exponential phase (*Figure 2C*). Consistent with these findings, a high-throughput CRISPRi study in *Sp* cells showed that TacL depletion led to increased lysis in stationary phase and aberrant cell morphology (*Liu et al., 2017*). Thus, *tacL* has the genetic properties expected for a gene encoding a LytA inhibitor that is active during normal exponential growth.

### TacL protects cells from extracellular LytA

LytA lacks a discernible motif for protein secretion and the mechanism by which it is exported has yet to be defined (*Díaz et al., 1989*). Therefore, one possible way in which TacL could control the ability of LytA to cleave the cell wall is through the inhibition of LytA secretion during exponential growth. Such a model predicts that both TacL$^+$ and TacL$^-$ cells should be equally sensitive to the addition of purified LytA. To test this possibility, recombinant LytA (rLytA) was purified from *Escherichia coli* (*Figure 3A*) and added to exponentially growing cultures of $\Delta lytA$ or $\Delta lytA$ $\Delta tacL$ cells (*Figure 3B*). As observed previously, addition of rLytA to $\Delta lytA$ (TacL$^+$) cells had no impact on growth during exponential phase and only caused lysis in stationary phase (*Figure 3B*) (*Fernebro et al., 2004*; *Mellroth et al., 2012*). However, the addition of rLytA to $\Delta lytA$ $\Delta tacL$ cells led to rapid cell lysis during exponential growth (*Figure 3B*). We therefore conclude that TacL is required for the growth-phase-dependent control of LytA activity at a step after its export to the cell surface.

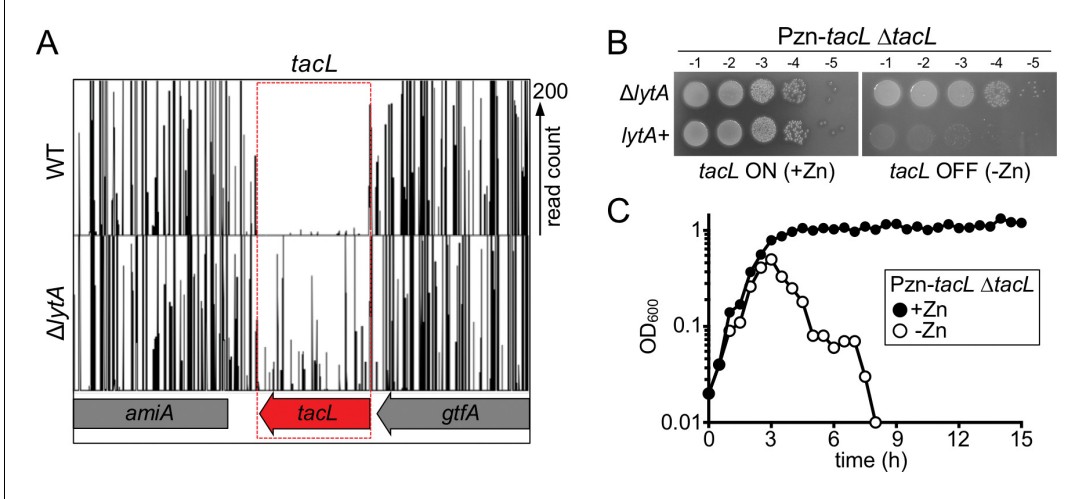

**Figure 2.** The essential gene *tacL* can be inactivated in cells lacking *lytA*. (**A**) Transposon insertion profiles from the Tn-Seq screen. Mariner transposon libraries were generated in wild-type (WT) and Δ*lytA* mutant strains and insertion sites were mapped to the *Sp* genome using Illumina sequencing. The height of each line reflects the number of sequencing reads at each position. Note that transposon insertions in *tacL* were much more readily isolated in cells lacking *lytA*. (**B**) Spot dilutions of the indicated strains in the presence and absence of inducer (Zn). The indicated strains were grown to exponential phase, normalized for $OD_{600}$ and serially diluted. Aliquots (5 µl) of each dilution were spotted onto TSAII 5% SB plates in the presence or absence of 100 µM $ZnCl_2$. Plates were incubated at 37 °C in 5% $CO_2$ and imaged. (**C**) Depletion of *tacL* results in growth arrest and lysis in exponential phase while its overexpression results in protection against growth-phase-dependent autolysis. Strains containing a zinc-inducible *tacL* allele (Pzn-*tacL*) were grown in THY to mid-exponential phase. Cultures were diluted into fresh THY to an $OD_{600}$ of 0.025 in the presence or absence of 100 µM $ZnCl_2$ and grown at 37 °C in 5% $CO_2$. Growth was monitored by taking $OD_{600}$ measurements approximately every 30 min for 15 hr.
DOI: https://doi.org/10.7554/eLife.44912.004

## TacL-dependent LTA biogenesis is antagonistic with WTA synthesis

TacL is a polytopic membrane protein predicted to have a large extracellular loop domain (*Figure 4A*). It was originally named RafX because the gene encoding it is located within a locus involved in raffinose utilization (*Wu et al., 2014*). Initial studies indicated that TacL was required for proper teichoic acid biogenesis (*Wu et al., 2014*), but its role in the process was not clear due the unique way in which *Sp* cells synthesize these polymers. Unlike most firmicutes, WTA and LTA in *Sp* cells have identical main chains (*Brown et al., 2013*; *Denapaite et al., 2012*; *Fischer et al., 1993*; *Gisch et al., 2013*; *Heß et al., 2017*; *Percy and Gründling, 2014*). Furthermore, bioinformatic analysis indicates that *Sp* cells are likely to make WTAs and LTAs from a common precursor polymer linked to an undecaprenyl phosphate (Und-P) lipid carrier (*Figure 4A*) (*Denapaite et al., 2012*). To make WTAs, the polymer is thought to be transferred to the cell wall by LCP-type enzymes (*Figure 4A*) (*Brown et al., 2013*; *Kawai et al., 2011*; *Percy and Gründling, 2014*; *Schaefer et al., 2017*). A recent study using mass spectrometry found that TacL is likely to be the corresponding LTA synthase (*Heß et al., 2017*). It is thought to be responsible for transferring the TA polymer from Und-P to the glycolipid anchor diglucosyl-diacylglycerol ($Glc_2$-DAG) to form LTAs (*Figure 4A*) and was therefore renamed TacL for teichoic acid ligase (*Heß et al., 2017*). Consistent with this assignment, HHPred analysis indicates that TacL shares remote similarity with O-antigen ligases (99.7% probability, E-value 5.1e-13) (*Heß et al., 2017*; *Wu et al., 2014*), enzymes from Gram-negative bacteria that carry out a similar reaction, the transfer of O-antigen glycan polymers from Und-P to a lipid A-core glycolipid acceptor (*Kalynych et al., 2014*).

We confirmed a role for TacL in LTA biogenesis by measuring LTA levels in exponentially growing cells with or without TacL inactivation (*Figure 4B*). Membrane preparations from whole cell lysates were analyzed by SDS-PAGE followed by immunoblotting with commercially available antibodies specific for the phosphatidyl-choline (PCho) modifications found on the TAs of *Sp* cells (*Denapaite et al., 2012*; *Fischer et al., 1993*; *Gisch et al., 2013*; *Percy and Gründling, 2014*). As reported previously using this method, we observed a ladder-like banding pattern of PCho-containing material ranging from 10 to 15 kDa in TacL[+] cells (*Figure 4B*) (*Wu et al., 2014*). As expected

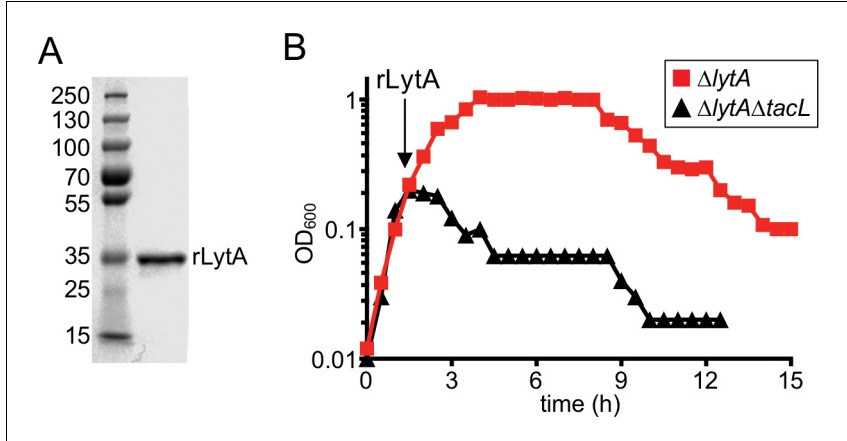

**Figure 3.** Cells lacking TacL are hypersensitive to exogenous LytA. (**A**) Coomassie-stained gel of recombinant LytA (rLytA) purified from *E. coli.* Molecular weight markers (in kDa) are shown. (**B**) Growth curves of the indicated strains before and after the addition of 1 µg/ml rLytA. Cells lacking *lytA* (Δ*lytA*) or *tacL* and *lytA* (Δ*lytA* Δ*tacL*) were challenged with rLytA at an $OD_{600}$ of ~0.2. In the absence of TacL, cells rapidly lyse after rLytA addition. By contrast and as reported previously, the Δ*lytA* (TacL[+]) strain only lyses in stationary phase in a manner similar to LytA[+] cells (*Fernebro et al., 2004*; *Mellroth et al., 2012*).

DOI: https://doi.org/10.7554/eLife.44912.005

based on the recent mass spectrometry study, the signal for this material was dramatically reduced in the Δ*tacL* strain and was restored upon complementation with *tacL* expressed from an ectopic locus (*Figure 4B*) (*Heß et al., 2017*). Thus, our results are consistent with the detected material indeed being LTA and that TacL is required for its formation. In a parallel set of samples, we measured the effect of TacL inactivation on the production of WTAs, which were detected as alcian blue-silver stained polymers released from purified cell wall sacculi (*Pollack and Neuhaus, 1994*). Only a modest level of WTA material was produced in exponentially growing TacL[+] cells (*Figure 4B*). However, TacL-defective cells had a striking increase in WTAs that was reduced back to near wild-type levels upon *tacL* complementation (*Figure 4B*). Our results thus provide additional support for the idea that TacL is the LTA ligase. Furthermore, the finding that WTAs increase when LTA synthesis is blocked suggests that the two pathways are antagonistic and are likely competing for the shared Und-P linked precursor (*Figure 4A*).

## A switch from LTA to WTA synthesis occurs at the onset of autolysis

Our results thus far indicate that loss of TacL function during exponential growth leads to a change in teichoic acid synthesis from LTA to WTA and the induction of LytA-dependent autolysis (*Figures 2*, *3* and *4*). Connecting these two phenomena is that LytA not only has a cell wall cleaving amidase domain, but also possesses an array of six choline-binding domains (CBDs) that promote its association with PCho-modified teichoic acids (*Fernández-Tornero et al., 2001*; *Li et al., 2015*; *Sandalova et al., 2016*). Thus, we reasoned that the LytA-dependent lethality observed upon TacL depletion may result from a switch in LytA localization from LTAs to WTAs where the enzyme will have better access to its substrate to promote cell wall degradation and lysis.

To test this hypothesis, we monitored the association of LytA with the membrane (LTA) and cell wall (WTA) in both TacL[+] and TacL[-] cells through different stages of growth. To avoid complications arising from autolysis, we used a catalytically inactive variant of LytA, LytA(H26A) to monitor its localization (*Mellroth et al., 2012*). However, to guide the collection of samples, particularly those corresponding to the onset of autolysis (A), we grew a WT (LytA[+]) strain alongside the LytA(H26A) producing cells (*Figure 5A*). In TacL[+] cells, we observed robust LTA production in exponential phase (E) and low levels of WTAs (*Figure 5A and B*). A significant portion of LytA(H26A) in these samples was found in the membrane fraction, presumably through association with LTAs (*Figure 5A and B*). The remaining portion of LytA was most likely retained in the cytoplasm awaiting export as has been reported previously (*Mellroth et al., 2012*). In early stationary phase (S), LTAs continued to be

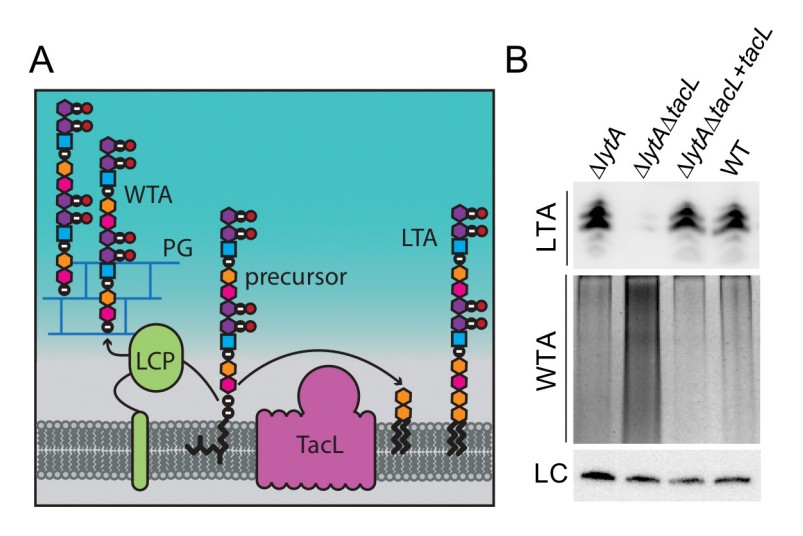

**Figure 4.** Cells lacking TacL contain altered levels of teichoic acids. (**A**) Schematic model depicting the final steps in the synthesis of WTA and LTA in *Sp*. WTAs and LTAs are thought to be synthesized from a common precursor polymer that is linked to an undecaprenyl phosphate lipid carrier (*Denapaite et al., 2012*). The polymer is transferred to the PG by LCP proteins to form WTA (*Denapaite et al., 2012*). TacL is hypothesized to transfer the precursor to a glycolipid anchor to generate LTA (*Heß et al., 2017*). (**B**) Analysis of LTA and WTA levels during exponential growth in wild-type (WT) or cells lacking LytA (Δ*lytA*), LytA and TacL (Δ*lytA*Δ*tacL*) or the double mutant harboring a Zn-inducible *tacL* allele (+*tacL*) grown in the presence of 100 μM ZnCl$_2$. Top: Immunoblot analysis of membrane-associated LTAs separated by 16% Tris-tricine SDS-PAGE and probed with a monoclonal antibody specific for phosphocholine. Middle: Analysis of WTAs released from purified cell wall sacculi and separated by SDS-PAGE followed by alcian blue-silver staining. Bottom: LC, loading control.
DOI: https://doi.org/10.7554/eLife.44912.006

detected and the amount of WTAs were observed to increase (*Figure 5A and B*). The majority of the extracellular LytA(H26A) remained associated with the membrane fraction, but a small amount was detected in the cell wall fraction (*Figure 5B*). Finally, as cells reached late stationary phase when autolysis normally begins, LTA levels were dramatically reduced while the amount of WTA had increased relative to the early stationary phase time point (*Figure 5A and B*). Now, LytA(H26A) was no longer found associated with the membrane fraction but instead its localization switched almost entirely to the cell wall fraction where it was presumably bound to the WTAs (*Figure 5A and B*). In similar experiments performed in Δ*tacL* cells, the WTAs were the only detectable teichoic acids at all growth stages, and LytA(H26A) was found associated with the cell wall fraction at every time point (*Figure 5C and D*).

We wondered whether the same change in teichoic acid production and LytA localization also occurred during penicillin-induced autolysis. Analogous to the previous set of experiments, a WT (LytA$^+$) strain was grown alongside a strain producing LytA(H26A) and served as a reference for the onset of lysis following penicillin-G (PenG) treatment (*Figure 6A*). Samples were harvested just before treatment (B), shortly after growth plateaued (P), and at the onset of lysis (L) (*Figure 6A and B*). We again saw a transition from LTA synthesis before treatment to the production of primarily WTAs at the onset of lysis (*Figure 6A and B*). Extracellular LytA(H26A) similarly was found to switch from the membrane to cell wall fraction during this time course (*Figure 6A and B*). We therefore conclude that both penicillin-induced and growth-phase-dependent autolysis follow a similar pathway involving a change from LTA to WTA synthesis and an accompanying transition in LytA localization. Based on these results, we infer that *Sp* cells control LytA activity during exponential growth by preferentially producing LTAs over WTAs to keep LytA at the membrane and prevent it from accessing its cell wall substrate. In response to prolonged stationary phase growth or treatment with penicillin, the synthetic bias switches. WTA production becomes favored, resulting in a change in LytA localization to the cell wall where it catalyzes the destruction of the matrix and promotes autolysis.

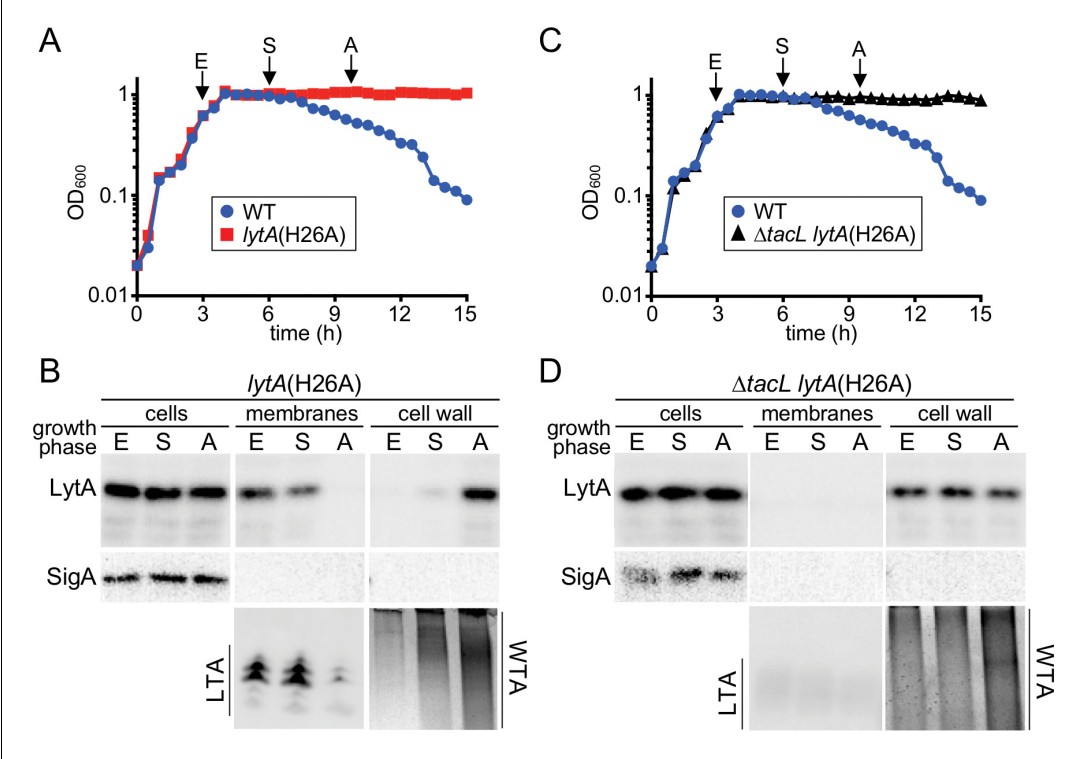

**Figure 5.** A switch from LTA to WTA synthesis and LytA localization occurs at the onset of autolysis. (**A and C**) Growth curves of the indicated strains cultured at 37 °C in 5% $CO_2$. At the indicated time points (E, exponential; S, stationary; A, autolysis), samples from the *lytA*(H26A) mutants were collected, normalized to an $OD_{600}$ of 0.5 and processed as described in Materials and methods. The WT growth curve was used as a reference for the timing of autolysis. (**B and D**) Samples from (**A**) and (**C**) were analyzed to detect LytA(H26A) in whole cell lysates (cells), associated with protoplast membranes, or cell wall sacculi. The cytoplasmic protein SigA was used as a control for protoplast integrity. The immunoblots were from the same membrane and exposure but were cropped to re-order the lanes for clarity. LTAs in the membrane fraction and WTA in the cell wall fraction were monitored by immunoblot and alcian blue-silver staining, respectively.

DOI: https://doi.org/10.7554/eLife.44912.007

## TAs are released from cells at the onset of autolysis

The rapid disappearance of LTAs at the onset of autolysis suggests that in addition to the switch from LTA production to WTAs, other mechanisms are at work to remove existing LTA molecules from the membrane. Accordingly, *Waks and Tomasz, 1978* previously observed the release of choline-containing TA polymers into the medium following treatment of *Sp* cells with cell wall synthesis inhibitors (*Waks and Tomasz, 1978*). To further investigate this phenomenon, cell membrane fractions and culture supernatants were harvested from LytA-defective cells before and during an autolysis time-course induced by either PenG treatment or stationary phase growth. As above, a WT (LytA+) strain was used as a reference to guide sample collection. In both cases, LTAs were detected in the membrane at early time points (E and B), but choline-containing material was absent from the supernatant (*Figure 7*). However, as autolysis progressed, an increase in choline-containing material in the supernatant was observed, while at late time points, LTAs in the membrane were barely detectable (*Figure 7*). Given their shared structure with WTAs (*Gisch et al., 2013*), it is not possible to definitively conclude that the released material is derived from LTAs. However, the coincidence of choline-containing polymers appearing in the medium with LTAs being lost from the membrane suggests that this material indeed reflects LTA release. Thus, in conjunction with the prior results of *Waks and Tomasz, 1978*, our results suggest that LTAs are released from cells during autolysis by an as yet unknown mechanism. Coupled with the switch in teichoic acid synthesis to favor WTAs over LTAs, LTA release would allow for the rapid depletion of LTAs and re-localization of LytA to WTAs where it can promote destruction of the wall matrix.

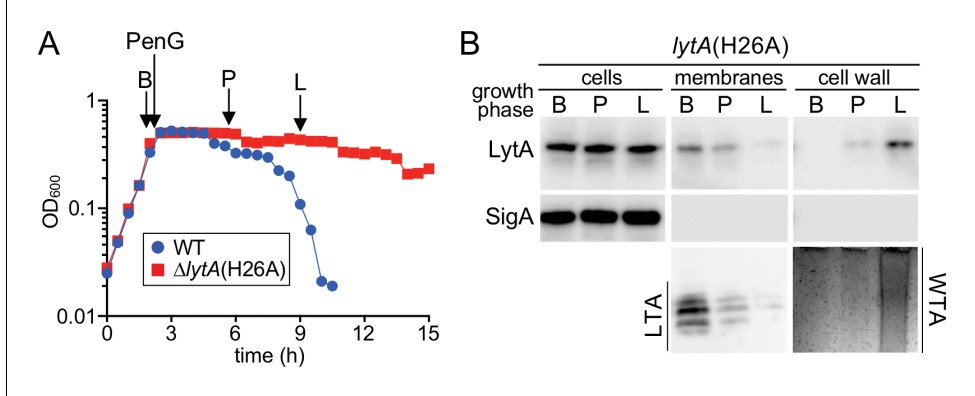

**Figure 6.** A switch from LTA to WTA synthesis and LytA localization occurs after exposure to penicillin. (**A**) Growth curves of the indicated strains before and after challenge with penicillin G (PenG) (0.5 μg/ml final). At the indicates time points (B, before PenG addition; A, after addition; L, lysis), samples from the *lytA*(H26A) strain were collected, normalized to an $OD_{600}$ of 0.5 and processed as described in Materials and methods. The growth curve of WT treated with PenG was used as reference for the timing of lysis (**B**) Samples from (**A**) were analyzed to detect LytA (H26A) in whole cell lysates (cells), associated with protoplast membranes, or with cell wall sacculi. The cytoplasmic protein SigA was used as a control for protoplast integrity. The immunoblots were from the same membrane and exposure but were cropped to re-order the lanes for clarity. LTAs in the membrane fraction and WTA in the cell wall fraction were monitored by immunoblot and alcian blue-silver staining, respectively.
DOI: https://doi.org/10.7554/eLife.44912.008

## The autolytic switch is triggered by FtsH-dependent degradation of TacL

We were next interested in determining how the switch in teichoic acid synthesis is initiated in response to autolytic conditions. Given that the change involves a loss of detectable LTAs, we suspected that TacL may be the primary regulatory target. As an initial test of this hypothesis, we monitored the effect of *tacL* overexpression ($P_{Zn}$-*tacL*) from an ectopic locus on autolysis induction. For both stationary-phase growth and penicillin treatment, *tacL* overexpression was found to prevent the induction of autolysis (*Figure 8—figure supplement 1*), suggesting that a reduction in TacL levels underlies the autolytic switch from LTA to WTA production.

To investigate this possibility further, a functional FLAG-tagged derivative of TacL was generated and its steady-state levels were monitored in Δ*lytA* cells at different phases of growth or following penicillin treatment (*Figure 8A and B*, and *Figure 8—figure supplement 2*). A reproducible decrease in TacL-FLAG abundance was observed at time points corresponding to autolysis onset in a reference LytA[+] strain (*Figure 8A and B*, and *Figure 8—figure supplement 2*). To determine whether this reduction in TacL-FLAG levels was caused by protease degradation, we monitored the half-life of TacL-FLAG following the inhibition of protein synthesis with chloramphenicol (Cm). The protein appeared to be relatively stable in exponential phase, but its half-life decreased significantly in late stationary phase or in response to penicillin treatment (*Figure 8C* and *Figure 8—figure supplement 2C*). We therefore conclude that TacL degradation is induced in response to conditions that trigger autolysis.

To identify the protease responsible for TacL degradation, we performed an additional Tn-seq screen. The rationale for the screen was that mutants unable to degrade TacL should be defective in the induction of autolysis and therefore more tolerant than WT cells to penicillin treatment. We used our original transposon library prepared in the WT strain and grew it either without drug or in the presence of a sub-lethal concentration of penicillin G that caused a mild growth defect without affecting viability. We then compared the transposon insertion profiles observed in the Tn-Seq data from these two conditions. A few genes were identified in which transposon insertions were significantly enriched in the penicillin treated condition versus the no drug or unrelated drug control (*Figure 8D*). One of these genes was *ftsH*, encoding a highly conserved ATP-dependent zinc metallopeptidase (*Ito and Akiyama, 2005*). FtsH has two transmembrane segments and a well-known

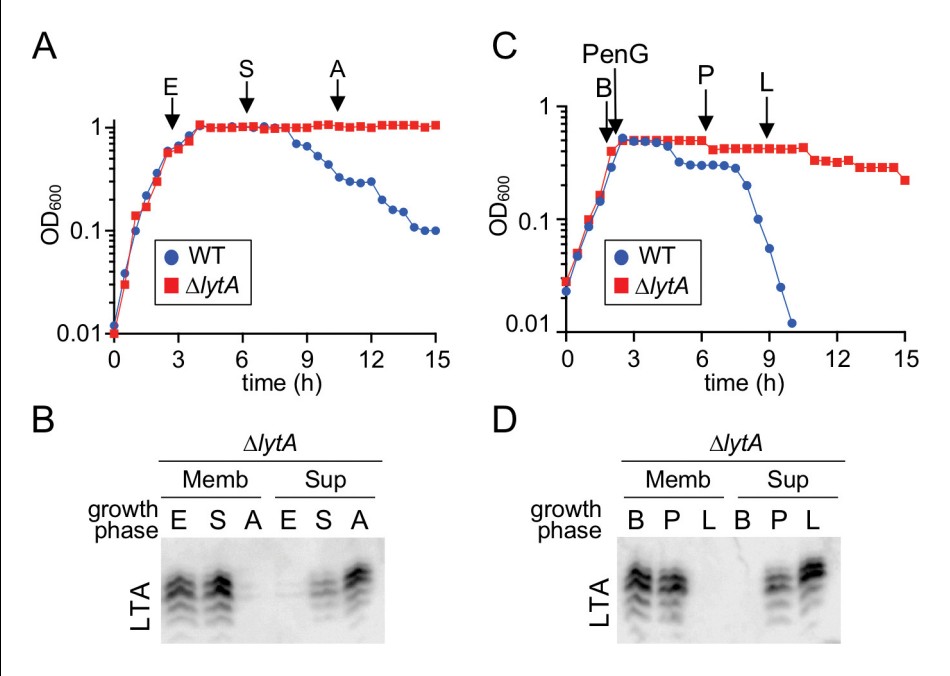

**Figure 7.** Teichoic acids are released into the culture medium during autolysis. (**A**) Growth curve of the indicated strains cultured at 37 °C in 5% $CO_2$. At the indicated time points (E, exponential; S, stationary; A, autolysis), samples from the $\Delta lytA$ mutant were collected, normalized to an $OD_{600}$ of 0.5 and processed as described in Materials and methods. The WT growth curve was used as a reference for the timing of autolysis. (**B**) Samples from (**A**) were analyzed to detect LTA in membranes (Memb) or culture medium (Sup, supernatant) by immonoblot. (**C**) Growth curve of the indicated strains cultured at 37 °C in 5% $CO_2$. At an $OD_{600}$ of ~0.5 the cultures were challenged with penicillin G (PenG) (0.5 µg/ml final). At the indicated time points (B, before PenG addition; A, after addition; L, lysis), samples from the $\Delta lytA$ mutant were collected and normalized to an $OD_{600}$ of 0.5. (**D**) Samples from (**C**) were analyzed to detect LTA in membranes (Memb) or supernatant (Sup) by immunoblot.
DOI: https://doi.org/10.7554/eLife.44912.009

role in the degradation of membrane protein substrates (*Langklotz et al., 2012*). We therefore focused on FtsH as a candidate for the protease targeting TacL.

To test whether FtsH targets TacL for degradation, we compared TacL-FLAG levels as well as LTA and WTA production in cells with or without FtsH. Unlike cells with functional FtsH, TacL-FLAG remained stable in $\Delta ftsH$ cells following penicillin treatment (*Figure 8A and B*). Moreover, these cells did not undergo the switch in teichoic acid production from LTAs to WTAs, nor did LytA[+] cells lacking FtsH autolyze in response to penicillin treatment (*Figure 8A and B*). Similar observations were made comparing FtsH[+] and FtsH[-] cells during prolonged stationary phase growth (*Figure 8—figure supplement 2A and B*). Specifically, TacL-FLAG remained stable in cells lacking FtsH and LTAs continued to be produced in stationary phase where their levels were dramatically reduced in FtsH[+] cells. Altogether, our results support a model in which autolysis in response to cell wall targeting drugs or stationary phase growth is caused by a switch in surface polymer biogenesis triggered by FtsH-mediated degradation of TacL.

## Discussion

Autolysis of *Sp* cells following entry into stationary phase was described by Walther Goebl and Oswald Avery almost a century ago (*Avery and Cullen, 1923*; *Goebel and Avery, 1929*; *Neufeld, 1900*). Although the biological relevance of this phenomenon remains unclear (*Eldholm et al., 2009*; *Hirst et al., 2008*; *Kietzman et al., 2016*; *Martner et al., 2009*), it is medically significant due to its relationship with antibiotic-induced bacteriolysis (*Fernebro et al., 2004*; *Mellroth et al., 2012*; *Tomasz et al., 1970*; *Tomasz and Waks, 1975*). The connection between these lytic events was

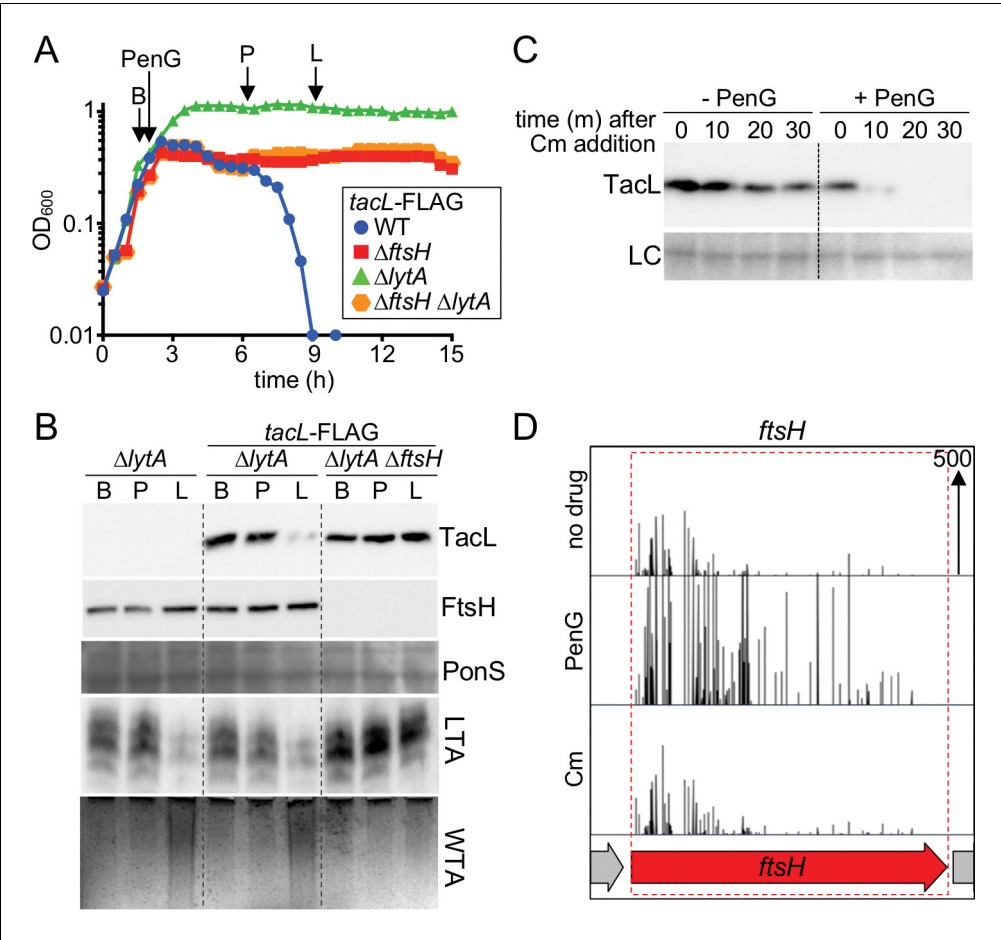

**Figure 8.** Penicillin treatment causes a reduction in TacL protein levels that depends on FtsH. (**A**) Growth curves of the indicated strains harboring a functional TacL-FLAG fusion as the sole source of TacL. At an $OD_{600}$ of ~0.5 the cultures were challenged with penicillin G (PenG) (0.5 μg/ml final). At the indicated time points (B, before penG addition; A, after addition; L, lysis), samples from the $\Delta lytA$ and $\Delta ftsH\Delta lytA$ strains were collected and normalized to an $OD_{600}$ of 0.5. Samples from a $TacL^+$ $\Delta lytA$ strain were collected at the same time points. The growth curve of this strain was omitted from the figure for clarity. The growth curve of the TacL-FLAG $LytA^+$ strain (WT) treated with PenG was used as reference for the timing of lysis. (**B**) Samples from (**A**) were processed as described in Materials and methods to detect TacL-FLAG and FtsH from whole cell lysates, LTA from membrane preparations, and WTAs released from purified cell wall sacculi. A region of the nitrocellulose membrane used for immunoblot analysis was stained with Ponceau S (PonS) to control for loading. (**C**) TacL-FLAG stability before and after PenG exposure. Wild-type *Sp* harboring TacL-FLAG was grown as in (**A**). Prior to PenG exposure (-PenG) or 2 hr after addition (+PenG), cultures were treated with chloramphenicol (Cm) (50 μg/ml final) to block translation. Samples were collected immediately before and 10, 20, 30 min after Cm addition and analyzed by SDS-PAGE and anti-FLAG immunoblotting to detect TacL. A region of the nitrocellulose membrane used for immunoblot analysis was stained with Ponceau S (PonS) to control for loading. (**D**) Transposon insertion profiles from a mariner transposon library generated in wild-type *Sp* and grown in the presence or absence of sub-inhibitory doses of penicillin G (PenG) or chloramphenicol (Cm). Transposon insertions in *ftsH* were significantly enriched (p<0.0001) in the presence of PenG compared to no drug or the Cm-treated control. The following figure supplements are available for **Figure 8**.

DOI: https://doi.org/10.7554/eLife.44912.010

The following figure supplements are available for figure 8:

**Figure supplement 1.** *tacL* overexpression prevents growth-phase-dependent and antibiotic-induced autolysis.
DOI: https://doi.org/10.7554/eLife.44912.011

**Figure supplement 2.** Reduction in TacL protein levels correlates with the growth phase-dependent switch from LTA to WTA synthesis and autolysis.
DOI: https://doi.org/10.7554/eLife.44912.012

made half a century later by Tomasz and co-workers (*Tomasz et al., 1970*; *Tomasz and Waks, 1975*). Their seminal studies demonstrated that both types of lytic events require the activity of the PG hydrolase LytA (*Tomasz et al., 1970*; *Tomasz and Waks, 1975*). Several decades have now elapsed since these discoveries were reported yet the regulatory systems controlling LytA during normal growth and how they are relieved to induce lysis have remained ill-defined. Here, we show that a change in cell surface polymer biogenesis underlies the onset of autolysis, providing a mechanistic framework to understand the process and exploit it for the development of novel lytic antibiotics.

Our data indicate that LTAs and WTAs have antagonistic effects on cellular LytA activity. LTAs are required to prevent LytA-induced lysis during exponential growth whereas an increase in WTA synthesis at the expense of LTAs is associated with conditions that promote cell wall destruction by LytA (*Figures 4*, *5*, *6*, *7* and *8*). The polymeric structure of LTA and WTA is identical in *Sp* cells (*Denapaite et al., 2012*; *Fischer et al., 1993*). LytA is therefore unlikely to have a different enzymatic potential when bound to these molecules through its choline-binding domains. Although it is possible that the association of LytA with WTA is required to properly orient the amidase domain with respect to its substrate for productive cleavage (*Li et al., 2015*; *Mellroth et al., 2014*; *Pérez-Dorado et al., 2010*; *Sandalova et al., 2016*), we favor a model in which the differential control exerted by these polymers is the result of how they affect the localization of LytA within the cell envelope (*Figure 9*).

Cryoelectron microscopy studies of Gram-positive bacteria by Beveridge and co-workers revealed that the PG layer is spatially separated from the cytoplasmic membrane, forming what has been referred to as the Gram-positive 'periplasm' (*Matias and Beveridge, 2008*; *Matias and Beveridge, 2005*; *Matias and Beveridge, 2006*). Evidence has also been presented that LTAs are likely to be restricted to this 'periplasmic' space (*Matias and Beveridge, 2008*; *Reichmann et al., 2014*). We therefore propose that in *Sp* cells, the protective function of LTAs is mediated by their ability to bind and sequester LytA in the membrane-proximal zone of the envelope thereby limiting its access to the cell wall and ability to cut bonds in the PG network (*Figure 9*). It is noteworthy that Tomasz and co-workers previously observed that purified LTAs were capable of inhibiting LytA-mediated autolysis by a mechanism that was unclear at the time (*Briese and Hakenbeck, 1985*; *Höltje and Tomasz, 1975*). Based on our findings, we infer that these results reflect the ability of micellar forms of LTAs to similarly bind and sequester LytA to prevent its association with cell wall substrate. After several hours in stationary phase or following penicillin treatment, we observed a dramatic reduction in cellular LTAs through the inhibition of their synthesis and what appears to be their release from the membrane (*Figure 7*). This drop in LTA levels is associated with a corresponding increase in the level of WTAs (*Figures 5*, *6* and *8*). At this time, LytA is similarly shifted from the membrane to cell wall fractions where it is likely bound to WTAs, its association with which has long been known to be required for cell wall cleavage activity (*Giudicelli and Tomasz, 1984*; *Höltje and Tomasz, 1975*; *Li et al., 2015*; *Mellroth et al., 2014*; *Tipper and Strominger, 1965*; *Tomasz et al., 1970*; *Waks and Tomasz, 1978*). Therefore, our results support a model in which it is the change in TA polymer synthesis and subsequent recruitment of LytA from membrane-anchored LTAs to its cell wall substrate that underlies autolysis induction (*Figure 9*).

Although we do not currently know the mechanism by which *Sp* cells preferentially make LTAs during exponential growth, our results establish that the switch to WTA synthesis under autolytic conditions involves the degradation of TacL, the likely ligase involved in making LTAs (*Figure 8* and *Figure 8—figure supplement 2*) (*Heß et al., 2017*). Furthermore, inactivation of the membrane-bound protease FtsH was shown to prevent TacL degradation and protect cells from LytA-dependent autolysis, suggesting that TacL is a direct substrate of FtsH (*Figure 8* and *Figure 8—figure supplement 2*). What remains to be determined is how TacL degradation by FtsH is triggered by conditions that induce autolysis.

The physiological signals governing FtsH activity are not known in any organism but are best understood in *E. coli*. As in *Sp* cells, *E. coli* FtsH controls an important branch point in an envelope biogenesis pathway (*Ogura et al., 1999*). In this case, its key substrate is LpxC, the enzyme catalyzing the committed step in the synthesis of lipopolysaccharide (LPS), a major component of the Gram-negative outer membrane (*Führer et al., 2006*; *Ogura et al., 1999*). The LPS and phospholipid synthesis pathways compete for fatty acids in the form of acyl-ACP precursors (*Mohan et al., 1994*; *Normark et al., 1969*). Therefore, flux between the two pathways must be balanced for

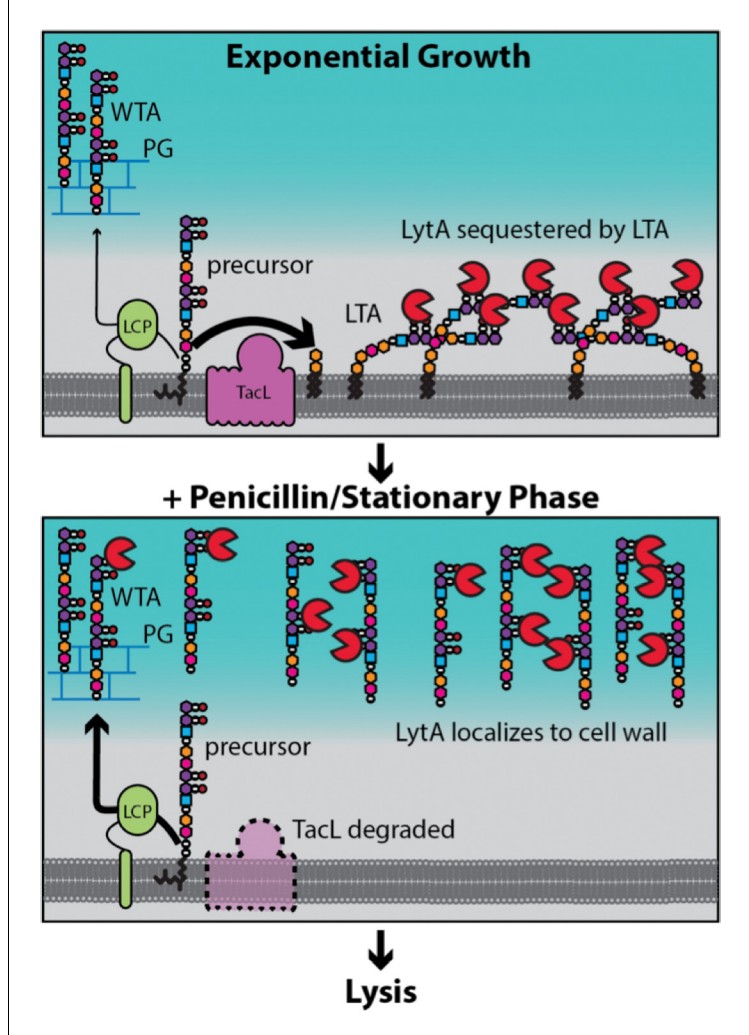

**Figure 9.** Model for LytA regulation. During exponential growth, TacL-dependent LTA synthesis dominates over LCP-dependent production of WTA. LytA bound to the phosphocholine moeities (purple/black balls) on LTA is sequestered away from WTA preventing LytA from targeting the cell wall. Upon entry into stationary phase or exposure to cell wall targeting antibiotics (penicillin), TacL is degraded in an FtsH-dependent manner leading to a reduction in LTA synthesis and an increase in WTA levels. Association of LytA with newly synthesized WTA leads to cell wall cleavage and lysis.

DOI: https://doi.org/10.7554/eLife.44912.013

proper membrane assembly. Indirect evidence suggests that the turnover of LpxC by *E. coli* FtsH may be regulated by the availability of certain acyl-ACP precursors or the accumulation of downstream products in the LPS synthesis pathway (*May and Silhavy, 2018*; *Mohan et al., 1994*; *Normark et al., 1969*; *Ogura et al., 1999*). Thus, in *Sp* cells, FtsH may similarly respond to precursors or products of the TA polymer assembly pathways it governs. As in *E. coli*, the sensing mechanism might normally function in the homeostatic control of LTA and WTA levels, which may be needed for more precise activation of LytA to promote septal PG cleavage and cell separation. Autolytic conditions would then be expected to somehow short-circuit the normal balancing mechanism, swinging it too far in favor of WTA synthesis such that destructive levels of LytA activity are stimulated. Further work will be required to test this and other possible models for the upstream signals governing *Sp* FtsH activity, their role in normal physiology, and how they are corrupted to induce autolysis.

TAs have been implicated in the control of PG hydrolases in other Gram-positive bacteria. In *Staphylococcus aureus*, the major autolysin Atl that acts at division septa to promote cell separation specifically localizes to the septal PG and this localization requires WTA (*Schlag et al., 2010*; *Yamada et al., 1996*). Interestingly, localization studies suggest that WTA levels are lowest at the septum in *S. aureus* suggesting that Atl preferentially localizes to PG lacking attached TA polymers (*Schlag et al., 2010*). In *B. subtilis*, the cell separase LytF localizes to septal PG in a manner that depends on both WTA and LTA (*Kiriyama et al., 2014*; *Yamamoto et al., 2008*). Finally, the major autolysin AcmA in *Lactococcus lactis* specifically localizes to the septum while galactosyl-containing TA polymers are largely absent at this site (*Steen et al., 2003*). In all of these cases the presence of TA polymers appears to interfere with autolysin binding to the cell wall, however the mechanisms underlying the temporal and spatial organization/localization of TAs is not known. Although it remains to be investigated, our results in *Sp* cells suggest that rather than one polymer or the other predominating, it may be the interplay between the LTA and WTA synthesis pathways that plays the key role in controlling PG hydrolysis and autolytic-induction in these and other Gram-positive organisms.

# Materials and methods

**Key resources table**

| Reagent type (species) or resource | Designation | Source or reference | Identifiers | Additional information |
|---|---|---|---|---|
| Strain (*Streptococcus pneumoniae*(*Sp*) D39 Δcps) | *WT (Sp D39 Δcps)* | Malcolm Winkler lab (*Lanie et al., 2007*) | Wildtype *S. pneumoniae* D39 Δcps | |
| Strain, strain background (*Sp* D39 Δcps) | AKF_Spn001 | *Fenton et al., 2016* | Δ*bgaA::kan* | |
| Strain, strain background (*Sp* D39 Δcps) | AKF_Spn002 | *Fenton et al., 2016* | Δ*bgaA::add9(spec)* | |
| Strain, strain background (*Sp* D39 Δcps) | AKF_Spn003 | *Fenton et al., 2016* | Δ*bgaA::tetM(tet)* | |
| Strain, strain background (*Sp* D39 Δcps) | AKF_Spn004 | *Fenton et al., 2016* | Δ*bgaA::cat* | |
| Strain, strain background (*Sp* D39 Δcps) | AKF_Spn005 | *Fenton et al., 2016* | Δ*bgaA::erm* | |
| Strain, strain background (*Sp* D39 Δcps) | AKF_Spn351 | *Fenton et al., 2016* | Δ*lytA::cat* | |
| Strain, strain background (*Sp* D39 Δcps) | AKF_Spn704 | This study | Δ*lytA::erm* | (see Materials and methods) |
| Strain, strain background (*Sp* D39 Δcps) | JFK_Spn001 | This study | Δ*lytA::erm* Δ*tacL::cat* | (see Materials and methods) |
| Strain, strain background (*Sp* D39 Δcps) | JFK_Spn002 | This study | Δ*ftsH::cat* | (see Materials and methods) |
| Strain, strain background (*Sp* D39 Δcps) | JFK_Spn003 | This study | Δ*lytA::erm* Δ*ftsH::cat* | (see Materials and methods) |
| Strain, strain background (*Sp* D39 Δcps) | JFK_Spn004 | This study | *lytA(H26A), erm* | (see Materials and methods) |

*Continued on next page*

*Continued*

| Reagent type (species) or resource | Designation | Source or reference | Identifiers | Additional information |
|---|---|---|---|---|
| Strain, strain background (*Sp D39 Δcps*) | JFK_Spn005 | This study | *lytA(H26A), erm ΔtacL::cat* | (see Materials and methods) |
| Strain, strain background (*Sp D39 Δcps*) | JFK_Spn006 | This study | *tacL-FLAG, spec* | (see Materials and methods) |
| Strain, strain background (*Sp D39 Δcps*) | JFK_Spn007 | This study | *tacL-FLAG, spec ΔftsH::cat* | (see Materials and methods) |
| Strain, strain background (*Sp D39 Δcps*) | JFK_Spn008 | This study | *tacL-FLAG, spec ΔlytA::erm* | (see Materials and methods) |
| Strain, strain background (*Sp D39 Δcps*) | JFK_Spn009 | This study | *tacL-FLAG, spec ΔlytA::erm ΔftsH::cat* | (see Materials and methods) |
| Strain, strain background (*Sp D39 Δcps*) | JFK_Spn010 | This study | *ΔbgaA::(P$_{Zn}$-tacL, tet)* | (see Materials and methods) |
| Strain, strain background (*Sp D39 Δcps*) | JFK_Spn011 | This study | *ΔbgaA::(P$_{Zn}$-tacL, tet) ΔlytA::erm* | (see Materials and methods) |
| Strain, strain background (*Sp D39 Δcps*) | JFK_Spn012 | This study | *ΔbgaA::(P$_{Zn}$-tacL, tet) ΔtacL::cat* | (see Materials and methods) |
| Strain, strain background (*Sp D39 Δcps*) | JFK_Spn013 | This study | *ΔbgaA::(P$_{Zn}$-tacL, tet) ΔtacL::cat ΔlytA::erm* | (see Materials and methods) |
| Strain, strain background (*E. coli DH5a*) | DH5a | Gibco BRL | *F-hsdR17 Δ(argF-lacZ)U169 phoA glnV44 Φ80dlacZ Δ M15 gyrA96 recA1 endA1 thi-1 supE44 deoR* | |
| Strain, strain background (*E. coli BL21*) | BL21 | New England Biolabs | *E. coli B F– ompT gal dcm lon hsdSB(rB–mB–) [malB+]K-12(λ S)* | |
| Recombinant DNA reagent | pLEM023 | *Fenton et al., 2016* | *bgaA'::P$_{zn}$::MCS:: tetM::bgaA' bla* | plasmid |
| Recombinant DNA reagent | pET24a | Novagen | *P$_{T7}$, lacI$^q$*; vector used for protein expression | plasmid |
| Recombinant DNA reagent | pMagellan6 | *Fenton et al., 2016* | *IRL(MmeI):: add9::IRR(MmeI) bla* | plasmid |
| Recombinant DNA reagent | 'pMalC9' | *Fenton et al., 2016* | *MBP::Himar1 bla* | plasmid |
| Recombinant DNA reagent | pJFK_001 | This study | *tacL* in pLEM023 | plasmid (see Materials and methods) |
| Recombinant DNA reagent | pJFK_002 | This study | *lytA* in pET24a | plasmid (see Materials and methods) |
| Antibody | LytA (rabbit polyclonal) | This study | | 1:50000 (see Materials and methods) |
| Antibody | FtsH (rabbit polyclonal) | *Kotschwar et al., 2004* | | 1:10000 |

*Continued on next page*

*Continued*

| Reagent type (species) or resource | Designation | Source or reference | Identifiers | Additional information |
|---|---|---|---|---|
| Antibody | SigA (rabbit polyclonal) | *Fujita, 2000* | | 1:10000 |
| Antibody | FLAG (TacL-FLAG; rabbit polyclonal) | Sigma | RRID:AB_796202 | 1:1000 |
| Antibody | LTA (mouse monoclonal) | Sigma | RRID:AB_1163630 | 1:1000; anti-Phosphocholine TEPC-15 |
| Antibody | anti-rabbit IgG-HRP | BioRad | RRID:AB_1102634 | 1:20000 |

## Strains, plasmids and routine growth conditions

All *Sp* strains were derived from the unencapsulated D39 strain (D39 $\Delta cps$) (*Lanie et al., 2007*). Cells were grown in Todd Hewitt broth (Beckton Dickinson) supplemented with 0.5% yeast extract (THY) at 37°C in an atmosphere containing 5% $CO_2$. Strains were grown on pre-poured tryptic soy agar 5% sheep blood plates (TSAII 5% sheep blood, Beckton Dickinson) with a 5 ml overlay of 1% nutrient broth (NB, Beckton Dickinson) agar containing the required additives or on TSA agar plates containing 5% defribrinated sheep blood (Northeast laboratory). Luria-Bertani (LB) broth and LB agar were used for *E. coli*. Antibiotic concentrations were used as previously described (*Fenton et al., 2016*). All strains, plasmids and oligonucleotides used in this study are provided in the key resources and the supplemental material 1, respectively. The D39 $\Delta cps$ genotype ($\Delta cps2A'$-$\Delta cps2H'$) was excluded from derivative strains for clarity. All *S. pneumoniae* strains used in this study are derivatives of D39 $\Delta cps$. Cam = chloramphenicol, Erm = erythromycin, Kan = kanamycin, Spec = spectinomycin, Tet = tetracycline, Amp = ampicillin.

## Transformation of *Sp*

Transformations were performed as previously described (*Fenton et al., 2016*). Briefly, cells in early exponential phase were back-diluted to an optical density at 600 nm ($OD_{600}$) of 0.03 and competence was induced with 500 pg/ml competence stimulating peptide 1 (CSP-1; Anaspec), 0.2% BSA, and 1 mM $CaCl_2$. Cells were transformed with 100 ng genomic DNA (gDNA) or plasmid DNA. Transformants were selected on TSAII overlay plates containing the appropriate additives.

## Growth curves

To monitor growth kinetics and autolysis, *Sp* cells in early exponential phase were diluted to an $OD_{600}$ of 0.025 and grown to mid exponential phase in THY media containing the appropriate additives at 37°C in an atmosphere containing 5% $CO_2$. These cells were used as the inoculum and were diluted to $OD_{600}$ of 0.025 in THY with the indicated additives and growth was monitored by measuring $OD_{600}$ every 30 min. The figures that report growth curves are representative of experiments that were performed on at least two independent samples.

## Library generation and transposon insertion sequencing (Tn-seq)

Tn-seq was performed as described previously (*Fenton et al., 2016*; *Fenton et al., 2018*). A total of two independently generated libraries were used in this study: one from D39 $\Delta cps$ (WT) and another from its $\Delta lytA$ derivative. Briefly, genomic DNA with Magellan6 transposon insertions generated in vitro was transformed into competent *Sp*. To ensure that more than 50% of all TA sites had at least one transposon insertion,>300,000 transformants were recovered from each library and gDNA isolated. gDNAs were digested with MmeI, followed by adapter ligation. Transposon–chromosome junctions were amplified and sequenced on the Illumina HiSeq. 2500 platform using TruSeq Small RNA reagents (Tufts University Core Facility Genomics; RRID:SCR_016383). Reads were demultiplexed, trimmed, and transposon- insertion sites mapped onto the D39 genome using methods described previously (*Fenton et al., 2016*; *Fenton et al., 2018*). After normalization, a Mann

Whitney U test was used to identify genomic regions with significant differences in transposon insertion profiles (*Fenton et al., 2016*; *Fenton et al., 2018*). Transposon insertion profiles were visualized using the Artemis genome browser (v10.2; RRID:SCR_004267) (*Carver et al., 2012*).

Additional libraries were also generated by replating the D39 $\Delta cps$ (WT) library in the presence and absence of 0.025 µg/ml penicillin G or 1.25 µg/ml chloroamphenicol. To maintain library complexity,>4,000,000 colonies were collected from each library and gDNA isolated and analyzed as described above.

## Isolation and analysis of pneumococcal LTAs

*Sp* strains were grown in THY at 37°C in an atmosphere containing 5% $CO_2$ to the indicated growth phase (additives were added as indicated) and normalized to an $OD_{600}$ of 0.5. 20 ml of the normalized culture was collected by centrifugation at 5000 g for 5 min and the cell pellet was washed twice with 2 ml SMM (0.5 M sucrose, 20 mM maleic acid, 20 $MgCl_2$, pH 6.5) before resuspending it in 2 ml SMM. Protoplasts were generated by addition of 20 mg/ml lysozyme and 100 units mutanolysin (Sigma) and incubated at 37°C for 30 min. Complete protoplasting was monitored by light microscopy. Protoplasts were pelleted by centrifugation at 5000 g for 5 min and resuspended in 2 ml cold hypotonic buffer to lyse them (20 mM HEPES ($Na^+$), pH 8.0, 100 mM NaCl, 1 mM dithiothreitol (DTT), 1 mM $MgCl_2$, 1 mM $CaCl_2$, 2X complete protease inhibitors (Roche), 6 µg/ml RNAse A, 6 µg/ml DNAse). Unbroken spheroplasts were removed by centrifugation 5,000 rpm for 10 min, and then the membrane fraction was collected by ultracentrifugation at 100,000 g for 1 hr at 4°C. The pellet was resuspended in 1 ml SDS sample buffer (200 mM Tris-HCL, pH 6.8, 40% glycerol, 2% SDS, 0.04% Coomassie Blue G-250), boiled for 10 min, and separated by Tris-tricine PAGE followed by immunoblotting with anti-PCho monoclonal antibody TEPC-15 (Sigma). To analyze TAs released into the culture medium, the supernatant from each sample was collected, flash-frozen using liquid nitrogen, and lyophilized. The lyophilized powder was then resuspended in 0.5 ml of distilled water. Samples were dialyzed against three 0.5 ml changes of distilled water followed by the addition of 0.5 ml of 2X SDS sample buffer (400 mM Tris-HCL, pH 6.8, 80% glycerol, 4% SDS, 0.08% Coomassie Blue G-250). To detect TAs, samples were analyzed as described above. The results in figures analyzing LTA levels are representative of experiments that were performed on at least two independently collected samples.

## Isolation and analysis of pneumococcal WTAs

*Sp* strains were grown in THY at 37°C in an atmosphere containing 5% $CO_2$ to the indicated growth phase and normalized to an $OD_{600}$ of 0.5. 20 ml of the normalized culture was collected by centrifugation at 7000 g for 5 min. The pellet was resuspended in 2 ml of buffer 1 (50 mM 2-(*N*-morpholino) ethanesulfonic acid (MES)), pH 6.5) and centrifuged at 7000 g for 5 min. The resulting pellet was resuspended in 2 ml of buffer 2 (50 mM MES, pH 6.5, 4% (w/v) SDS) and incubated in boiling water for 1 hr. After incubation, the cell wall sacculi were collected at 7000 g for 5 min and washed with 2 ml of buffer 2. The sample was transferred into a clean microfuge tube and centrifuged at 16000 rpm for 5 min. The pellet was then washed with 2 ml of buffer 2, followed by successive washes with 2 ml of buffer 3 (50 mM MES, pH 6.5, 2% (w/v) NaCl) and 2 ml of buffer 1. The samples were then centrifuged at 16000 rpm for 5 min, resuspended in 2 ml of buffer 4 (20 mM Tris-HCl, pH 8.0, 0.5% (w/v) SDS) supplemented with 2 µl of proteinase K (20 mg/ml), and incubated at 50°C for 4 hr with shaking (1000 rpm). After incubation, the pellet was collected by centrifugation and washed with 2 ml of buffer 3 followed by three washes with distilled water. The pellet was then collected by centrifugation and subjected to alkaline hydrolysis by resuspending the pellet in 0.5 ml of 1N sodium hydroxide solution and incubation at 25°C for 16 hr with shaking (1000 rpm). Insoluble cell wall material was pelleted by centrifugation (13000 rpm for 5 min) and the supernatants containing the extractable WTA were collected and combined 1:1 with 0.5 ml native sample buffer (62.5 mM Tris-HCl, pH 6.8, 40% glycerol, 0.01% bromophenol blue). To detect WTA, samples were analyzed by native PAGE followed by alcian-blue silver staining as described in Pollack *et al.* (*Pollack and Neuhaus, 1994*). The figures that report WTA levels are representative of experiments that were performed on at least two independently collected samples.

## LytA subcellular fractionation

*Sp* strains were grown in THY at 37°C in an atmosphere containing 5% $CO_2$ to the indicated growth phase (PenG was added as indicated) and normalized to an $OD_{600}$ of 0.5. 40 ml of the normalized culture was collected by centrifugation at 5000 g for 5 min. The cell pellet was washed twice with 2 ml SMM followed by resuspension with 2 ml of SMM and divided into two 1 ml samples (Samples A and B) and pelleted by centrifugation. Sample A (whole cell lysate) was lysed by resuspension in 500 µl of lysis buffer (see below), followed by addition of 500 µl SDS sample buffer containing 10% 2-mercaptoethanol and boiled for 10 min. To determine the amount of LytA that localized with the cell wall (WTA-bound) and the membrane (LTA-bound) fractions, sample B was resuspended in 500 µl SMM and protoplasts were generated by the addition of 20 mg/ml lysozyme and 100 units muta-nolysin (Sigma) and incubated at 37°C for 30 min. Complete protoplasting was monitored by light microscopy. The protoplasts were then pelleted (Sample C) and the supernatant containing the cell wall (Sample D) was collected. 500 µl of SDS-sample buffer containing 10% 2-mercaptoethanol was then added to sample D and boiled for 10 min to release WTA-bound LytA. To release LytA from membrane-bound LTA, sample C was incubated with 500 µl SMM supplemented with 2% choline chloride (w/v) for 30 min at 25°C with gentle shaking. The protoplasts were then pelleted by centrifugation, and the supernatant containing LytA was collected. 500 µl of SDS-sample buffer containing 10% 2-mercaptoethanol was added to the supernatant fraction and then boiled for 10 min. Samples A, C, and D were analyzed by SDS-PAGE followed by immunoblotting with anti-LytA and anti-sigA antisera. WTAs in sample D were analyzed by alcian-silver blue staining. LTAs in sample C were analyzed by immunoblot using anti-PCho monoclonal antibody TEPC-15 (Sigma). The figures that report LytA levels are representative of experiments that were performed on at least two independently collected samples.

## Measurement of TacL-FLAG steady-state level

*Sp* strains were grown in THY at 37°C in an atmosphere containing 5% $CO_2$ to the indicated growth phase (PenG was added as indicated). Cultures were normalized to an $OD_{600}$ of 0.5 and 5 ml of each were harvested immediately before and 10, 20, 30 min after addition of chloramphenicol (50 µg/ml final concentration) to block translation. The cultures were maintained at 37°C for the duration of the experiment. Samples were then analyzed by SDS-PAGE and immunoblotting using anti-FLAG poly-clonal antibody (Sigma). The figures that report TacL-FLAG levels are representative of experiments that were performed on at least two independently collected samples.

## Recombinant LytA (rLytA) purification

Recombinant LytA was produced in *E. coli* BL21(DE3) containing the pET24a-*lytA* expression vector. Cells were grown in LB supplemented with 50 ug/ml kanamycin at 37°C with vigorous agitation and *lytA* expression was induced when cells reached an $OD_{600}$ of 0.5 with 1 mM IPTG for 2 hr at 37°C. Cells were collected by centrifugation and stored overnight at −20°C. The cell pellets were resus-pended in *E. coli* lysis buffer (20 mM Tris, pH 7.5, 500 mM NaCl, DNase 200 µg/ml, and 2X complete protease inhibitors (Roche)). The cell suspension was then lysed by two passages through a cell dis-ruptor (Constant systems Ltd.) at 25000 psi and unbroken cells were removed by centrifugation. The supernatant was then passed over a DEAE cellulose column (Sigma). After washing with 20 column volumes of wash buffer (20 mM $NaPO_4$, 1.5M NaCl, pH 7), LytA was eluted with two column volumes of wash buffer supplemented with 140 mM choline chloride. Protein-containing fractions were pooled and dialyzed against 20 mM Tris, pH 7.5, 150 mM NaCl, 5 mM choline chloride and 10% glycerol and flash-frozen in liquid $N_2$ and stored at −80 °C.

## Antisera and immunoblot analysis

*Sp* cultures were normalized to an $OD_{600}$ of 0.5 and 5 ml harvested by centrifugation. Cell pellets were resuspended in 100 ul of lysis buffer (20 mM Tris pH 7.5, 10 mM EDTA, 1 mg/ml lysozyme, 100 units mutanolysin (Sigma), 10 µg/ml DNase I, 100 µg/ml RNase A, and 2X complete protease inhibi-tors (Roche Applied Sciences) and incubation at 37°C for 10 min. Equal volume of SDS sample buffer (100 µl, 0.25 M Tris pH 6.8, 4% SDS, 20% glycerol, 10 mM EDTA) containing 10% 2-mercaptoethanol was added. Proteins were separated by SDS-PAGE, electroblotted onto nitrocellulose membrane and blocked in 5% nonfat milk in phosphate-buffered saline (PBS)−0.5% Tween-20. The blocked

membranes were probed with anti-LytA (1:50,000), *B. subtilis* anti-FtsH (1:10,000) (*Kotschwar et al., 2004*), *B. subtilis* anti-SigA (*Fujita, 2000*) (1,10,000), anti-FLAG (Sigma; RRID:AB:796202) (1:1,000), and anti-PCho TEPC-15 (Sigma; RRID:AB_1163630) (1:1,000) diluted into 1% nonfat milk in 1X PBS-0.05% Tween-20. Primary antibodies were detected using horseradish peroxidase-conjugated goat, anti-rabbit IgG (1:20,000, BioRad; RRDID:AB_1102634) and the Western Lightning reagent kit as described by the manufacturer (PerkinElmer). Chemiluminescence was imaged on a FluorChem R system (ProteinSimple).

## Strain construction

### *Sp* deletion strains

All *Sp* deletion strains were generated using linear PCR fragments as described in *Fenton et al. (2016)* are listed in *Supplementary file 1* (*Fenton et al., 2016*). Briefly, regions of approximately 1 kb flanking each gene were amplified, and an antibiotic resistance cassette placed between them using isothermal assembly. Assembled PCR products were transformed directly into *Sp* as described above. In all cases, deletion primers were given the name: 'gene name'_5FLANK_F/R for 5′ regions and 'gene name'_3FLANK_F/R for 3′ regions. Antibiotic markers were amplified from Δ*bgaA::antibiotic cassette* (*bga* gene disrupted with an antibiotic cassette) strains using the AB_Marker_F/R primers. A full list of primer sequences can be found in the *Supplementary file 1*. Extracted gDNA from deletion strains was confirmed by diagnostic PCR using the AntibioticMarker_R primer in conjunction with a primer binding ~200 bp 5′ of the disrupted gene; these primers were given the typical name: 'gene name'_Seq_F. Confirmed gDNAs of single gene deletions were diluted to 20 ng/µl and used for the construction of multiple knockout strains. For strains containing multiple deletions and construct integrations, transformants were verified by diagnostic re-streaking on media containing the proper antibiotics. When needed, each construct was confirmed by diagnostic PCR and/or sequencing.

### P$_{zn}$-*tacL*

The *tacL* ORF, with its native RBS, was amplified using primers *tacL_F_nativeRBS_XhoI* and *tacL_R_BamHI*. The primers introduced XhoI and BamHI sites used for cloning into pLEM023 cut with the same enzymes, resulting in plasmid pJFK001. The plasmid was sequenced and used to transform strain D39 Δ*cps* Δ*bga::kan*. Integration into the *bga* locus was confirmed by antibiotic marker replacement and PCR using the BgaA_FLANK_F primer. gDNA from the resulting strain was prepared and then used to transform the appropriate *Sp* strains.

### tacL-FLAG

The *tacL* ORF, including its native promoter and RBS and a C-terminal FLAG sequence, was generated by isothermal assembly from 3 PCR products: 1) a PCR product containing an upstream region of *tacL* and the *tacL* ORF (including a C-terminal FLAG sequence) amplified with oligos *tacL_5F_F* and *tacL_FLAG*; 2) a PCR product containing a Spec cassette amplified with oligos AB_Marker_FLAG_F and AB_Marker_R; and 3) a PCR product containing a downstream region of *tacL* amplified with oligos *tacL_3F_F* and *tacL_3F_R*. The assembled product was used to transform strain D39 Δ*cps* Δ*lytA::erm* Δ*tacL::cat*. Integration into the Δ*tacL::cat* locus was confirmed by antibiotic marker replacement and PCR, sequencing, and immunoblot analysis. gDNA from the resulting strain was prepared and then used to transform the appropriate *S. pneumoniae* strains.

### lytA-H26A

The *lytA* ORF, including its native promoter and RBS and the H26A mutation, was generated by isothermal from 4 PCR products: 1) a PCR product containing an upstream region of *lytA* and the H26A mutation amplified with oligos LytA_5F_F and LytA_H26A_R; 2) a PCR product containing the LytA-H26A mutation and the 3′ end of *lytA* using oligos LytA_H26A_F and LytA_ABMarkerF_R; 3) a erythromycin cassette amplified with oligos AB_Marker_ F and AB_Marker_R; and 4) a PCR product containing a downstream region of *lytA* amplified with oligos *lytA_3F_F_AB_Marker_R* and *lytA_3F_R*. The assembled product was used to transform strain D39 Δ*cps* Δ*lytA::erm.* Integration into the Δ*lytA::cat* locus was confirmed by antibiotic marker replacement and PCR, sequencing, and

immunoblot analysis. gDNA from the resulting strain was prepared and then used to transform the appropriate *Sp* strains. *pET24a-lytA (pJFK002)*

The *lytA* ORF was amplified using primers *lytA_F_purification_NdeI* and *lytA_R_purification_HindIII.* The primers introduced NdeI and HindIII sites used for cloning into pET24a cut with the same enzymes, resulting in plasmid pJFK002. The plasmid was confirmed by sequencing.

## Acknowledgements

The authors would like to thank all members of the Bernhardt and Rudner laboratories for support and helpful comments. We also thank Michael Welsh for helpful insights regarding the isolation of LTAs and WTAs, Neil Greene and Mary Nahorniak for help with the growth and manipulation of *Sp*, Lok-To Sham for help in the purification of rLytA, David Tracy for technical help, and Malcolm Winkler for strains. We also thank Thomas Wiegert for providing the anti-FtsH antibody. JFK is a jointly mentored postdoctoral fellow bridging work in both the Bernhardt and Rudner labs. This work was supported by the Howard Hughes Medical Institute (TGB) and the National Institutes of Health (R01AI083365 to TGB, CETR U19 AI109764 to TGB and DZR, GM073831 to DZR, RC2 GM092616 to DZR, R01AI139083 to TGB and DZR, and F32AI36431 to JFK).

## Additional information

### Funding

| Funder | Grant reference number | Author |
| --- | --- | --- |
| National Institutes of Health | F32AI36431 | Josué Flores-Kim |
| National Institutes of Health | U19 AI109764 | David Z Rudner<br>Thomas G Bernhardt |
| National Institutes of Health | R01 GM073831 | David Z Rudner |
| National Institutes of Health | RC2 GM092616 | David Z Rudner |
| National Institutes of Health | R01AI139083 | David Z Rudner<br>Thomas G Bernhardt |
| Howard Hughes Medical Institute | | Thomas G Bernhardt |
| National Institutes of Health | R01 AI083365 | Thomas G Bernhardt |

The funders had no role in study design, data collection and interpretation, or the decision to submit the work for publication.

### Author contributions

Josué Flores-Kim, Conceptualization, Investigation, Writing—original draft, Writing—review and editing; Genevieve S Dobihal, Investigation; Andrew Fenton, Resources; David Z Rudner, Conceptualization, Supervision, Funding acquisition, Investigation, Writing—original draft, Project administration, Writing—review and editing; Thomas G Bernhardt, Conceptualization, Supervision, Funding acquisition, Writing—original draft, Project administration, Writing—review and editing

### Author ORCIDs

Josué Flores-Kim http://orcid.org/0000-0001-8282-6647
David Z Rudner https://orcid.org/0000-0002-0236-7143
Thomas G Bernhardt http://orcid.org/0000-0003-3566-7756

### Decision letter and Author response

Decision letter https://doi.org/10.7554/eLife.44912.017
Author response https://doi.org/10.7554/eLife.44912.018

## Additional files

**Supplementary files**

• Supplementary file 1. Table of oligonucleotides used in this study

DOI: https://doi.org/10.7554/eLife.44912.014

• Transparent reporting form

DOI: https://doi.org/10.7554/eLife.44912.015

**Data availability**

All data generated or analyzed during this study are included in the manuscript and supporting files.

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
