## [Decision Letter]

Thank you for submitting your article "A switch in surface polymer biogenesis triggers growth-phase-dependent and antibiotic-induced bacteriolysis" for consideration by *eLife*. Your article has been reviewed by two peer reviewers, and the evaluation has been overseen by a Reviewing Editor and Gisela Storz as the Senior Editor. The following individuals involved in review of your submission have agreed to reveal their identity: Jan Willem Veening (Reviewer #2).

The reviewers have discussed the reviews with one another and the Reviewing Editor has drafted this decision to help you prepare a revised submission.

In this interesting paper, Flores-Kim and co-workers come up with an exciting model for how LytA (and potentially other autolysins) becomes activated. They show that inactivation of TacL (aka as waaL, rafX, spr1708, SP_1893 or SPD_1672), the Lipo teichoic acid (LTA) synthase, increases the levels of wall TAs and leads to LytA-dependent lysis. This suggests that LTAs sequester LytA away from the cell wall. As LTAs are mainly made during exponential growth cells are protected from LytA autolytic activity. The authors also identify FtsH as the main protease responsible for degradation of TacL during stationary phase or when cells get treated with Penicillin G. How this exactly works is not yet clear. This work provides several new insights into the long-standing question on how LytA activity is controlled. In addition, it provides some leads as to the regulation of autolysins in other bacteria via controlled production of LTA and WTA.

Essential Revisions:

Additional verifications to improve the paper are listed below:

1) Activation by WTAs. According to the model, LytA needs WTA for its activation. Since LTA and WTA share the same precursors, depleting for instance tarP/tarQ or repressing the lic1 locus (tarI) (see e.g. Liu et al., 2017) should result in cells with less WTA (and less LTA), thus making them less susceptible to rLytA.

2) Link between LytA and LTA release.

– the authors show that TacL is the crucial enzyme regulating the switch from LTA to WTA

– the switch is controlled by the FtsH integral membrane protease degradation of TacL

This is consistent with data generated by Tomasz and colleagues that observed that lysis could be blocked by inhibiting protein synthesis with chloramphenicol and that this inhibition was not related to LytA abundance. However, Tomasz and colleagues also showed that lysis was triggered as a consequence of release of LTA into the supernatant. So, although TacL and FtsH seem important, their identification does not explain the entire process because even though TacL is degraded, if the pre-synthesized LTA remains on the cell membrane, LytA should stay sequestered at the cell membrane and prevent lysis. So, what is triggering LTA release? Is TacL involved in this process?

The authors have followed the production of LTA and WTA on the bacteria but have not monitored the released LTA and WTA. Do they also observe that lysis is linked to LTA release in the *tacL* mutant? If TacL is involved in the release, then the TacL overproducing strain should not release any LTA either in stationary phase or after penicillin treatment. Alternatively, LTA might be still released but compensated by the overexpression of TacL leading to enough LTA on the cell membrane to sequester LytA and block lysis. In the latter case, it would mean that something else is responsible for the LTA release and that this is dependent on protein synthesis as shown by Tomasz et al. Testing this would be essential to close the final gap.

Other suggested modifications:

– Subsection “A switch from LTA to WTA synthesis occurs at the onset of autolysis”, last paragraph. This conclusion appears a bit premature on the basis of Figure 6. In Figure 6 PenG was added at very late exponential phase (looking at the graph OD 0.4-0.5)? It would be better to disentangle PenG's effects from the 'natural' switch in LTA to WTA if this experiment was performed by adding PenG at mid-exponential phase (i.e. OD 0.1-0.2).

– Subsection “Identification of TacL as a potential LytA control factor”. The correct reference for P_Zn_ is Eberhardt et al., 2009. It is correct that Kloosterman et al. demonstrated that PczcD is activated by Zn^2+^, but it was Eberhardt et al. who made the connection to utilize this promoter as an inducible promoter for gene depletion studies.

– Subsection "Identification of TacL as a potential LytA control factor”. Liu et al., 2017 showed the effects of a *tacL* (spd_1672) depletion by CRISPRi and reported increased lysis in stationary phase. Cells deleted for *tacL* were also chained with heterogenous cell shapes. These data seem in line with the data reported here and it would be appropriate to mention. However, apparently, it is possible to make a *tacL* deletion in encapsulated LytA^+^ D39 bacteria. In addition, there seem to be some Tn-insertions in *tacL* even in wild type cells (Figure 2A). Can the authors comment on this?

– Discussion, first paragraph. An interesting hypothesis recently put forward by Kietzman et al., 2016, is that LytA is required for shedding of the capsule, thereby enabling adherence to epithelial cells.

– Subsection “Library generation and transposon insertion sequencing (Tn-seq)”, last paragraph. Chloroamphenicol – Figure 2A, Figure 7D, it is not clear why total reads at that location is plotted. It would be clearer to plot unique transposon insertion positions as spurious amplification can enrich certain sites.

– Figure 2A legend. This seems an incorrect statement as there are some insertions visible.

---

## [Author Response]

Essential Revisions:Additional verifications to improve the paper are listed below:1) Activation by WTAs. According to the model, LytA needs WTA for its activation. Since LTA and WTA share the same precursors, depleting for instance tarP/tarQ or repressing the lic1 locus (tarI) (see e.g. Liu et al., 2017) should result in cells with less WTA (and less LTA), thus making them less susceptible to rLytA.

The reviewers are correct. In theory, depleting factors required for the biogenesis TA precursors should reduce cellular levels of WTA and LTA simultaneously. Thus, according to our model, with reduced WTAs for LytA to associate with, cells should likely be more resistant to LytA autolysis. However, the TA biogenesis genes are essential, and in our experience depletion strains that block WTA synthesis become quite sick and grow poorly under depletion conditions. Therefore, in practice, the proposed experiment is one in which the results will likely be difficult to interpret (and hard to control for) due to the sickly state of the cells being studied. Nevertheless, a choline-binding requirement for LytA activity is well supported by the literature such that the proposed experiment is not necessary to support our model.

It is well-established that LytA is a choline binding protein that requires association with choline-modified TA polymers to degrade the cell wall and promote autolysis (see Tomasz, Albino and Zanati, 1970; Giudicelli and Tomasz, 1984; Severin et al., 1997). Combining our results that LTAs are protective against autolysis with historical studies showing a choline-binding requirement for LytA action, we think it is reasonable to conclude that WTAs are required for LytA-induced autolysis. We have added this line of reasoning to our overview of the model in the Discussion section (third paragraph).

2) Link between LytA and LTA release.– the authors show that TacL is the crucial enzyme regulating the switch from LTA to WTA– the switch is controlled by the FtsH integral membrane protease degradation of TacLThis is consistent with data generated by Tomasz and colleagues that observed that lysis could be blocked by inhibiting protein synthesis with chloramphenicol and that this inhibition was not related to LytA abundance. However, Tomasz and colleagues also showed that lysis was triggered as a consequence of release of LTA into the supernatant. So, although TacL and FtsH seem important, their identification does not explain the entire process because even though TacL is degraded, if the pre-synthesized LTA remains on the cell membrane, LytA should stay sequestered at the cell membrane and prevent lysis. So, what is triggering LTA release? Is TacL involved in this process?

*The authors have followed the production of LTA and WTA on the bacteria but have not monitored the released LTA and WTA. Do they also observe that lysis is linked to LTA release in the* tacL *mutant? If TacL is involved in the release, then the TacL overproducing strain should not release any LTA either in stationary phase or after penicillin treatment. Alternatively, LTA might be still released but compensated by the overexpression of TacL leading to enough LTA on the cell membrane to sequester LytA and block lysis. In the latter case, it would mean that something else is responsible for the LTA release and that this is dependent on protein synthesis as shown by Tomasz et al. Testing this would be essential to close the final gap.*

The reviewers raise an excellent point. The degradation of TacL to block new LTA synthesis is clearly an important component of the autolytic process, but it is also likely to involve the release of existing LTAs. We have assayed for released TA material in the supernatant during stationary-phase and penicillin-induced autolysis. We indeed find released choline-containing material in the supernatant of cultures, and the appearance of this released material is coincident with the loss of LTAs. Given that WTAs and LTAs share the same polymeric structure, we cannot conclude with certainty that the released material is LTA. However, the timing of its appearance is consistent with the material being LTA released during autolysis. TacL does not appear to be involved in the release because it is degraded at the time release is observed. These new experiments are now described in the Results section (subsection “TAs are released from cells at the onset of autolysis”) and presented in Figure 7, and the notion that LTAs are likely being released as part of the autolytic process has been incorporated into our discussion of the model (Discussion, third paragraph). We are actively searching for the factor responsible for LTA release, but its identification is beyond the scope of the current report.

Other suggested modifications:– Subsection “A switch from LTA to WTA synthesis occurs at the onset of autolysis”, last paragraph. This conclusion appears a bit premature on the basis of Figure 6. In Figure 6 PenG was added at very late exponential phase (looking at the graph OD 0.4-0.5)? It would be better to disentangle PenG's effects from the 'natural' switch in LTA to WTA if this experiment was performed by adding PenG at mid-exponential phase (i.e. OD 0.1-0.2).

We agree that it would be ideal to perform the PenG addition experiment earlier in exponential phase. However, the experiment was performed as reported so that we had enough cell material for detecting LytA and the TAs in the various fractions. Although the graphs look similar due to the semi-log scale, we never see stationary-phase autolysis until cells reach an OD_600_ > 1. The response we see from PenG addition at OD_600_ = 0.4-0.5 is immediate such that we think it is safe to conclude that the observed effects are the result of PenG addition and not stationary-phase.

– Subsection “Identification of TacL as a potential LytA control factor”. The correct reference for P_Zn_ is Eberhardt et al., 2009. It is correct that Kloosterman et al. demonstrated that PczcD is activated by Zn^2+^, but it was Eberhardt et al. who made the connection to utilize this promoter as an inducible promoter for gene depletion studies.

Thank you for pointing this out. We have changed the reference as suggested (subsection “Identification of TacL as a potential LytA control factor”).

– Subsection "Identification of TacL as a potential LytA control factor”. Liu et al., 2017 showed the effects of a tacL (spd_1672) depletion by CRISPRi and reported increased lysis in stationary phase. Cells deleted for tacL were also chained with heterogenous cell shapes. These data seem in line with the data reported here and it would be appropriate to mention. However, apparently, it is possible to make a tacL deletion in encapsulated LytA^+^ D39 bacteria. In addition, there seem to be some Tn-insertions in tacL even in wild type cells (Figure 2A). Can the authors comment on this?

We have added the indicated reference to Liu et al., 2017 in the Results section (subsection “Identification of TacL as a potential LytA control factor”). We have similarly observed a heterogeneous morphology for TacL^-^ defective cells.

We have found *tacL* to be essential in D39 strains regardless of capsule status. In D39 Cps^+^ cells, transforming a Δ*tacL* allele yields 1000-fold fewer transformants in a LytA^+^ background versus a LytA^-^ strain. No transformants were isolated in a D39 Cps^-^ cells. We suspect the recovered *tacL* deletion the reviewer is referring to is likely to have a suppressor mutation that inactivates *lytA*. Similarly, the very small number of insertions detected in *tacL* in the Tn-Seq profile of WT cells in Figure 2A are likely either due to spurious PCR amplification or instances of cells with suppressing mutations. It is rare to have a completely clean Tn-Seq profile for an essential gene, even those encoding factors like DNA polymerase that cannot be made non-essential by secondary mutations.

– Discussion, first paragraph. An interesting hypothesis recently put forward by Kietzman et al., 2016, is that LytA is required for shedding of the capsule, thereby enabling adherence to epithelial cells.

Thank you for pointing this out. We were aware of this interesting hypothesis. However, it is one of many hypotheses in the literature providing a possible physiological role for autolysis. Because the data in the literature do not overwhelmingly support one hypothesis over another, we state that the physiological significance remains unclear, but now include references to several papers suggesting potential functions for autolysis.

– Subsection “Library generation and transposon insertion sequencing (Tn-seq)”, last paragraph. Chloroamphenicol – Figure 2A, Figure 7D, it is not clear why total reads at that location is plotted. It would be clearer to plot unique transposon insertion positions as spurious amplification can enrich certain sites.

We agree that spurious amplification can lead to random insertion sites being enriched in a profile. This possibility is why we like to represent the Tn-Seq data as shown in Figures 2A and 7D. With these profiles, all unique Tn insertion sites detected are indeed plotted (location along the x-axis) along with their abundance in the population (the height of the line). Therefore, if one particular insertion site were the result of spurious amplification, it would easily be visualized as an outlier (a single line in the profile with a large number of reads).

– Figure 2A legend. This seems an incorrect statement as there are some insertions visible.

We have modified the wording to state, “Note that transposon insertions in *tacL* were more readily isolated in cells lacking *lytA*.”

References:

Severin A., Horne, D., Tomasz A., (1997) Autolysis and Cell Wall Degradation in a Choline-Independent Strain of *Streptococcus pneumoniae.* Microb Drug Resist 3:391-400